# Risk of HBV reactivation in HBV/HCV-co-infected HCV-treated patients: A single-center study

**Young Joo Park, Ki Youn Yi, Hyun Young Woo*, Jeong Heo*, Geun Am Song**

Department of Internal Medicine, College of Medicine, Pusan National University and Biomedical Research Institute, Pusan National University Hospital, Busan, Republic of Korea

☯ These authors contributed equally to this work.
* jheo@pusan.ac.kr (JH) and who54@hanmail.net (HYW)

## Abstract

Hepatitis B virus (HBV) reactivation in patients with HBV/hepatitis C virus (HCV) co-infection due to direct-acting antiviral agent (DAA) therapy is a growing concern. This study focused on 47 patients with chronic hepatitis C (CHC) and positivity for HBV surface antigen (HBsAg) who were treated with interferon (IFN)-based therapy, DAA, or DAA after IFN-based therapy failure and followed for a median of 53 months. Here, we aimed to determine HBV reactivation rates and associated factors, the incidence of HBV and liver-related events, and the rate of sustained virologic response (SVR) for HCV. Fifteen (15/47, 31.9%) patients experienced HBV reactivation during or after HCV treatment. This reactivation occurred significantly more frequently in patients who received DAA treatment after IFN-based treatment failure than in those who received IFN-based treatment (IFN-based vs. DAA vs. DAA treatment after IFN-based treatment failure 11.8% vs. 35.3% vs. 53.8%, respectively; p = 0.046). The interval from HCV treatment initiation to HBV reactivation was shortest in the DAA group (4.2 months), followed by the DAA after IFN-based treatment failure group (6.4 months) and the IFN-based treatment group (44.5 months) (p < 0.001). One case of HBV-related hepatitis spontaneously resolved after 4 weeks. The rate of SVR for the entire cohort was 87.2%, with no significant difference in this regard among the IFN-based treated, DAA-treated, and DAA-treated after IFN-based treatment failure arms at 82.4%, 88.2%, and 92.3%, respectively. HBV reactivation in HBsAg-positive CHC patients is more common and occurs earlier in those who receive DAA treatment after IFN-based treatment failure than in those with IFN-based treatment. Therefore, all patients with CHC should be tested for HBV exposure prior to DAA treatment. In addition, HBsAg positive patients, especially those among whom have previously experienced IFN-based treatment failure, should be closely monitored for HBV reactivation during DAA therapy.

**Data availability statement:** All relevant data are within the paper and its Supporting Information files.

**Funding:** The author(s) received no specific funding for this work.

**Competing interests:** The authors have declared that no competing interests exist.

## Introduction

Hepatitis B virus (HBV) and hepatitis C virus (HCV) co-infection is fairly common due to the similar modes of transmission of these two viruses [1,2]. The prevalence of HBV and HCV co-infection in Korea was reported to be 1.5% to 2.4% [3,4], which is lower than the rate of 1%−15% estimated worldwide [5,6]. This discrepancy in HBV/HCV co-infection prevalence can be attributed to varying population demographics, differing co-infection rates with other viruses, and the influence of testing methods, which may underestimate co-infection in untreated patients. There are three clinical presentations of HBV/HCV co-infection: acute hepatitis due to simultaneous infection with HBV and HCV, hepatitis B or C superinfection in patients with pre-existing chronic viral hepatitis, and occult HBV infection in patients with HCV infection [7]. In regions with a high prevalence of HBV infection (such as countries in the Asia-Pacific region), HCV superinfection in patients with chronic hepatitis B (CHB) is the most clinical form of HBV/HCV co-infection [8–10].

HBV reactivation involves the sudden reappearance or increase in HBV DNA serum levels in patients with a previously resolved or inactive HBV infection. It is most commonly triggered by immunosuppressive therapies such as cancer chemotherapy, corticosteroids, and other immunosuppressive drugs [11]. The reactivation rate has been reported to range from 3% to 55%, particularly with rituximab in HBsAg-positive patients [12], and is approximately 12% in non-liver solid organ transplant recipients with a history of prior HBV infection [13]. When HCV is effectively cleared through treatment, it can alter immune regulation in treated patients and increase the risk of HBV reactivation [14,15]. HBV reactivation after interferon (IFN)-based hepatitis C treatment occurred in 19.1%–36.4% of patients [16,17]. The US Food and Drug Administration has identified reports of HBV reactivation in 24 cases [including 9 patients with detectable HBV and 7 HBV surface antigen (HBsAg)-positive patients] who were receiving direct-acting antivirals (DAAs), which are more effective than interferon in eradicating HCV, with two cases resulting in death and one necessitating liver transplantation [18].

In the published literature on HBV reactivation after the treatment of hepatitis C in HBV/HCV-co-infected patients, clinically significant HBV reactivation events have been reported on rare occasions [19–22]. However, the available reports are mostly case studies and are of limited value in answering questions about HBV reactivation during DAA treatment in HBV/HCV-co-infected patients encountered in routine clinical practice. To shed more light on this issue, we evaluated a retrospective cohort of consecutive HBsAg-positive chronic hepatitis C (CHC) patients administered an IFN-based treatment, DAAs, or both in routine clinical practice.

## Methods

### Patients

CHC was diagnosed when laboratory tests were seropositive for HCV antibodies and HCV RNA for more than 6 months. Between January 2005 and June 2022, 2,330 patients were treated for CHC at a tertiary hospital in Republic of (South) Korea. Of

these, 66 of 73 individuals with evidence of HBV exposure at screening were included in this retrospective cohort study, excluding 7 patients with occult HBV infection (HBsAg-negative, positive for antibodies to HBV core antigen). Of these 66 HBsAg-positive patients, 47 were analyzed, excluding 19 [10 who were already taking nucleoside analogues (NUC) at the start of hepatitis C treatment, 7 who were not treated for hepatitis C, and 2 who were lost to follow-up] (Fig 1). The study was approved by Pusan National University Hospital Institutional Review Board (IRB, approval No. 2112-018-110) and conducted in accordance with the principles of the Declaration of Helsinki and the International Conference on Harmonization for Good Clinical Practice. All patients provided their informed consent for hepatitis C treatment, and the retrospective design of this study was approved by the IRB, which exempted the need for a formal consent form. All data were accessed for research from the date of IRB approval (December 24, 2021) to June 30, 2022.

CHC, chronic hepatitis C; HBV, hepatitis B virus; HCV, hepatitis C virus; NUC, nucleoside analogues; IFN, interferon; DAA, direct-acting antiviral.

## Treatment protocol

Before the introduction of DAAs in 2015, patients were treated with IFN-based therapy. Patients in this cohort received either IFN-based treatment or DAA treatment, and additional DAA treatment was administered if a sustained virological response (SVR) was not achieved with IFN-based treatment since the introduction of DAA in 2015. Pan-genotypic DAAs, such as glecaprevir/pibrentasvir, were available in Korea in September 2018. Therefore, the most appropriate DAA treatment regimen was selected based on the guidelines for the treatment of hepatitis C released by the Korean Association for the Study of the Liver.

## Assessments

Laboratory tests, including HBV serology, HBV DNA, HCV RNA, HCV genotype, complete blood counts, and liver and renal function tests were performed before treatment. HBV DNA, HCV RNA, complete blood counts, and liver and renal function tests were performed after 4 weeks of treatment, at the end of treatment, and 12 weeks after the end of treatment (EOT). Serum alpha-fetoprotein (AFP) measurement and liver ultrasonography were performed for hepatocellular

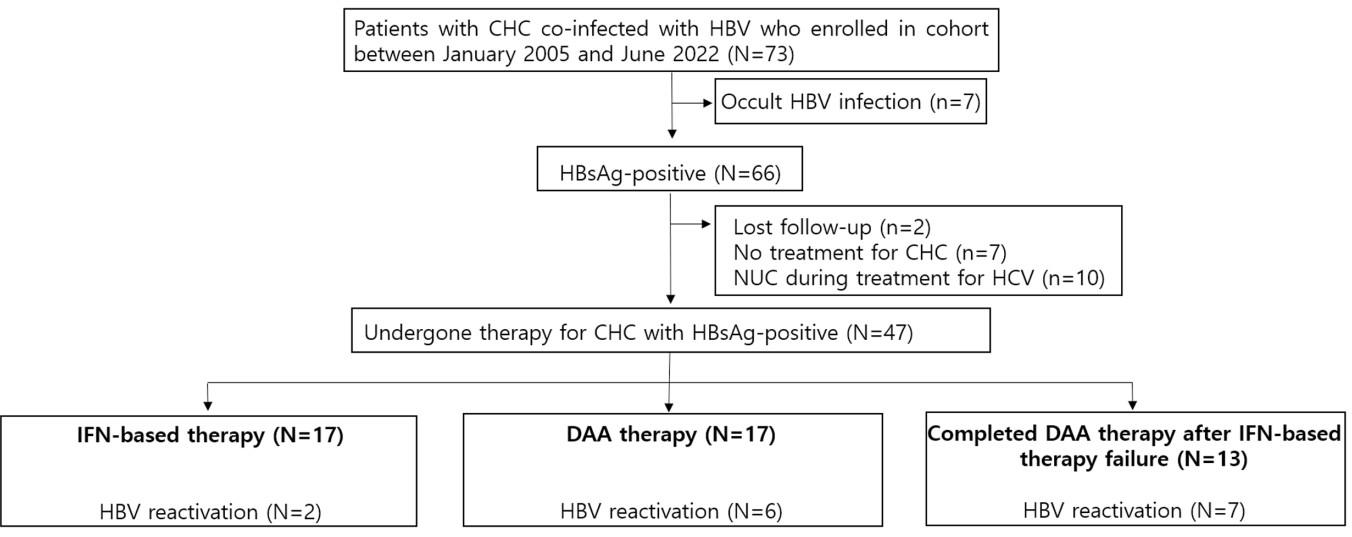

**Fig 1. The patient flow-chart.**

carcinoma (HCC) surveillance every 6 months. At each visit, vital signs were recorded and a physical examination was performed. Moreover, the occurrence of adverse events was checked and recorded in the patient's medical records.

HBsAg, antibody to hepatitis B surface antigen (anti-HBs), hepatitis B envelope antigen (HBeAg), and antibody to hepatitis B envelope antigen (anti-HBe) were tested using a radioimmunoassay (Abbott Laboratories, Abbott Park, IL, USA). HBV DNA was measured using a real-time PCR (RT-PCR) assay on a Cobas TaqMan 48 Analyzer (Roche Molecular Diagnostics, Branchburg, NJ, USA), which had a detection limit of 20 IU/mL. Serum HCV RNA levels and HCV genotype were measured by RT-PCR (COBAS TaqMan Analyzer; Roche Molecular Systems Inc., Pleasanton, CA, USA), with a lower quantitative detection limit of 15 IU/mL; no detection and concentrations below 15 IU/mL were reported separately.

### Endpoints

The primary endpoint of the study was HBV reactivation. HBV reactivation was defined as an increase of at least 1 $\log_{10}$ in the HBV DNA level in patients with detectable HBV DNA before treatment or HBV DNA detection in patients who had undetectable HBV DNA levels before treatment. Secondary outcomes included HBV-related hepatitis, efficacy of HCV therapy, and liver-related events. HBV-related hepatitis is generally defined as evidence of HBV reactivation plus an increase in alanine aminotransferase (ALT), usually more than 2–5 times the baseline value or upper limit of normal (ULN) [23,24]. SVR 12 weeks after the EOT (SVR12) was used to assess the efficacy of HCV therapy.

### Statistical analysis

All categorical variables are reported as counts and percentages and were compared using Pearson's chi-square test or Fisher's exact test. All continuous variables were reported as medians and ranges and were compared using the Mann–Whitney U-test. Univariate and multivariate logistic regression analyses were used to identify factors significantly associated with HBV reactivation. All statistical tests were two-sided, and p-values of <0.05 were considered significant. Statistical analyses were conducted using SPSS version 22.0 (IBM Corp., Armonk, NY, USA).

## Results

### Clinical characteristics of HBsAg-positive CHC patients (Table 1)

Of the 47 patients, 57.4% were male, and 57.4% were infected with HCV genotype 1. In addition, 36.2% of them received IFN-based treatment, 36.2% received DAA treatment, and 27.6% received DAA treatment after IFN-based treatment failure. The age at HCV treatment was lowest in the IFN-based treatment group, followed by the DAA-treated group after IFN-based treatment failure and then the DAA-treated group; the IFN-based treatment group was significantly younger than the DAA-treated group after IFN-based treatment failure (p = 0.007). There were no statistically significant genotype differences between the DAA groups with and without IFN-based treatment failure. After IFN-based treatment failure, sofosbuvir plus ribavirin was the most common treatment regimen in the DAA group, whereas daclatasvir plus asunaprevir was the most common treatment regimen in the DAA group (p = 0.033).

### Efficacy of HCV treatment in HBV/HCV-co-infected patients

Of the 47 patients included in the analysis, 87.2% reached HCV SVR12 (Fig 2). The SVR12 was 82.4% (14/17) in the IFN-treated group, 88.2% (15/17) in the DAA-treated group, and 92.3% (12/13) in the DAA-treated group after IFN treatment failure. The SVR12 for different DAA treatment regimens were as follows: the daclatasvir and asunaprevir combination group achieved 87.5% (7/8), the sofosbuvir and ribavirin combination group achieved 71.4% (5/7), and all other DAA treatment regimens achieved 100% (S1 Fig), which were not significantly different.

SVR, sustained virological response; HCV, hepatitis C virus; HBV, hepatitis C virus; SVR12, SVR 12 weeks after the end of treatment; IFN, interferon; DAA, direct acting antiviral.

## Clinical outcomes of HBV infection among HBV/HCV-co-infected patients

**HBV reactivation.** During a median follow-up period of 53 months, 31.9% (15/47) of the patients experienced HBV reactivation during or after HCV treatment (Fig 3). HBV reactivation occurred in 11.8% (2/17) of IFN-based therapy patients, 35.3% (6/17) of DAA-treated patients, and 53.8% (7/13) of DAA-treated patients after IFN treatment failure (p = 0.046).

The characteristics of the 15 patients who experienced HBV reactivation during or after HCV treatment are shown in Table 2. The median interval from HCV treatment initiation to HBV reactivation was shortest in the DAA group (4.2 months), followed by the DAA after IFN treatment failure group (6.4 months) and the IFN group (44.5months). Specifically, the interval was significantly shorter in both the DAA treatment group and the DAA after IFN treatment failure group than in the IFN group (both p < 0.001). However, there was no significant difference in this regard between the DAA treatment group and the DAA after IFN treatment failure group.

In 73% (11/15) of HBV reactivations, HBV DNA that was undetectable before treatment became detectable during or after treatment. Four (26.7%) patients with HBV reactivation had detectable HBV DNA before HCV treatment, with two of them having an increase of 1 log10 and the other two having an increase of at least 2 log10. After HBV reactivation, HBV DNA increased to a median of 807 IU/mL and decreased to a median of 19 IU/mL 8 months after HBV reactivation.

**HBV-related hepatitis.** One patient had HBV-related hepatitis according to this study's definition. HBV DNA was undetectable before HCV treatment but increased to 41 IU/mL and ALT to 113 U/mL 12 weeks after DAA initiation. After 4 weeks of HBV reactivation, the HBV DNA level had fallen below the lower limit of detection [20], and the ALT level had normalized and was followed up without NUC. The other patient was prescribed NUC after HBV reactivation because

**Table 1. Clinical characteristics of HBsAg-positive CHC patients.**

| Characteristics | IFN treatment N = 17 | DAA treatment N = 17 | DAA after failure of IFN treatment N = 13 | Total N = 47 |
|---|---|---|---|---|
| Age at HCV treatment, years, median (ranges) | 57 (28-71) | 70 (42-80) | 59 (48-73) | 61 (28-80) |
| Male sex, n (%) | 11 (64.7) | 8 (47.1) | 8 (61.5) | 27 (57.4) |
| Liver cirrhosis, n (%) | 4 (23.5) | 5 (29.4) | 6 (46.2) | 15 (31.9) |
| HCV genotypes, n (%) 1 2 3 4 Unknown | 7 (41.2) 6 (35.3) 1 (5.9) 1 (5.9) 2 (11.8) | 9 (52.9) 8 (47.1) 0 (0.0) 0 (0.0) 0 (0.0) | 11 (84.6) 2 (15.4) 0 (0.0) 0 (0.0) 0 (0.0) | 27 (57.4) 16 (34.0) 1 (2.1) 1 (2.1) 2 (4.3) |
| History of HCC | 0 (0.0) | 2 (11.8) | 3 (23.1) | 5 (10.6) |
| DAA treatment regimen DCV + ASV SOF + RBV DCV + SOF OPr-D G/P SOF + LDV | | 3 (17.6) 7 (41.2) 0 (0.0) 2 (11.8) 5 (29.4) 0 (0.0) | 5 (38.5) 0 (0.0) 1 (7.7) 2 (15.4) 3 (23.1) 2 (15.4) | 8 (26.7) 7 (23.3) 1 (3.3) 4 (13.3) 8 (26.7) 2 (6.7) |
| Detectable HBV DNA before treatment | 10 (58.8) | 3 (17.6) | 3 (23.1) | 16 (34.0) |

HCV, hepatitis C virus; HCC, hepatocellular carcinoma; DAA, direct acting antiviral; DCV, daclatasvir; ASV, asunaprevir; SOF, sofosbuvir; RBV, ribavirin; OPr-D, ombitavir/paritaprevir/ritonavir plus dasabuvir; G/P, glecaprevir/pibrentasvir; LDV, ledipasvir; IFN, interferon.

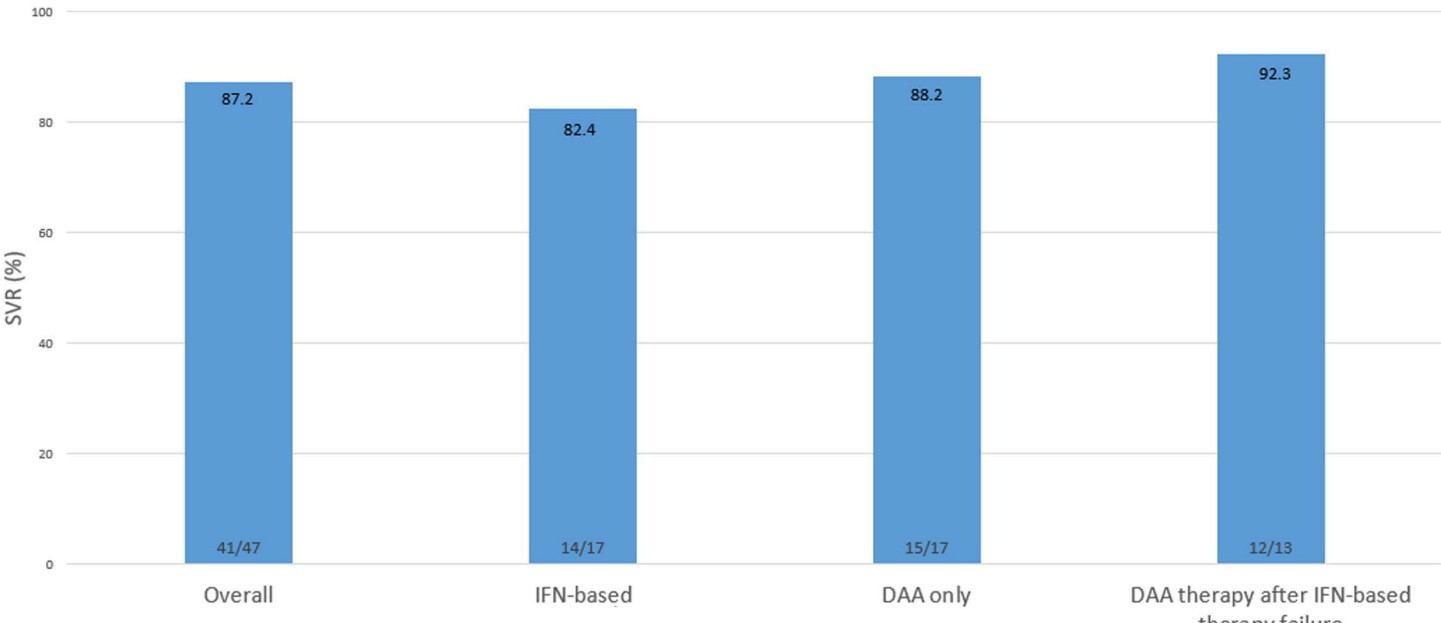

**Fig 2. SVR12 of HCV treatment in HBV/HCV-co-infected patients.**

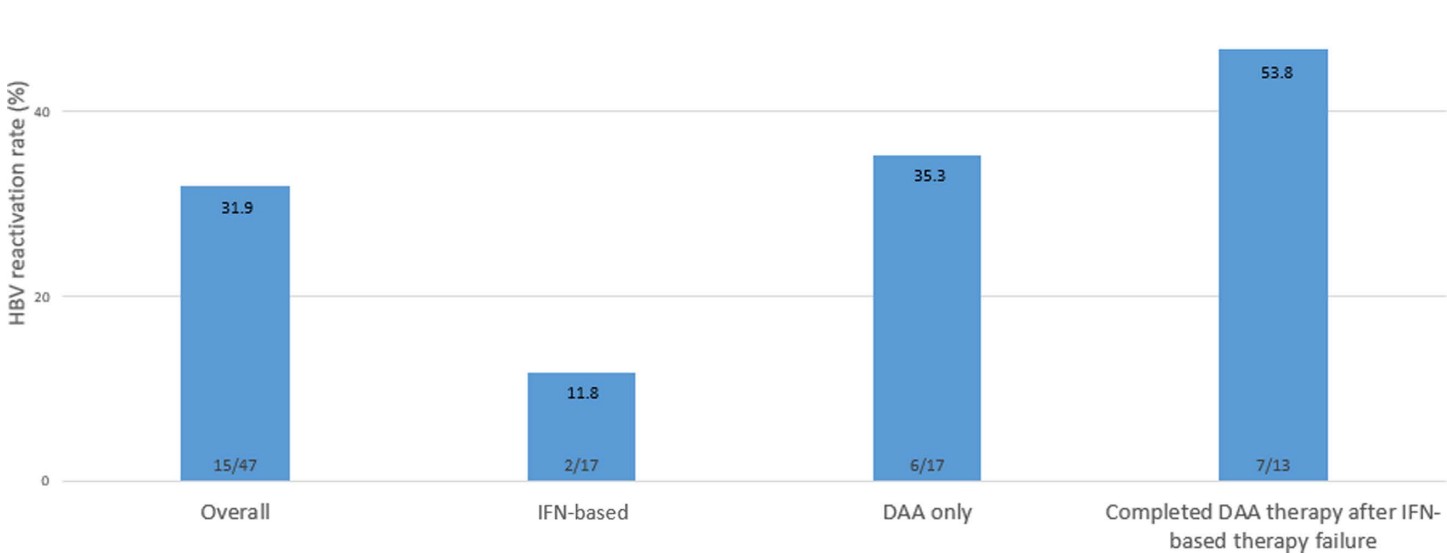

**Fig 3. HBV reactivation rate during or after therapy for HCV according to the treatment group.** HBV, hepatitis B virus; HCV, hepatitis C virus; IFN, interferon; DAA, direct acting antiviral.

**Table 2. Virological characteristics of patients with HBV reactivation after antiviral therapy for HCV.**

| Case | Sex | Age | HCV GT | LC | HCV treatment | HCV SVR | Pretreatment | | At HBV reactivation | | | | | |
|------|-----|-----|--------|-----|---------------|---------|--------------|---|---------------------|---|---|---|---|---|
| | | | | | | | HCV RNA (IU/mL) | HBV DNA (IU/mL) | Time from HCV treatment start date (weeks) | HBV DNA (IU/mL) | ALT (IU/mL) | Peak HBV DNA (IU/mL) | Peak ALT (IU/mL) | NUC ther-apy |
| 1 | M | 28 | 3a | No | IFN+RBV | Yes | 1,320,000 | 1.3 | 173 | 1,510 | 13 | 2,360 | 35 | No |
| 2 | F | 52 | 1b | Yes | IFN+RBV | Yes | 3,140,000 | Not detected | 219 | 526 | 18 | 778 | 38 | No |
| 3 | F | 80 | 1b | No | DCV+ASV, 24 weeks | Yes | 62,000 | 41.8 | 4 | 7,480 | 17 | 7,480 | 39 | No |
| 4 | F | 76 | 1b | No | OPr-D, 12 weeks | Yes | 504,000 | Not detected | 12 | 835 | 23 | 835 | 23 | No |
| 5 | F | 69 | 2a/2c | Yes | SOF+RBV, 16 weeks | Yes | 1,890,000 | Not detected | 54 | 35 | 14 | 35 | 15 | No |
| 6 | F | 63 | 1b | No | DCV+ASV, 24 weeks | Yes | 1,390,000 | Not detected | 12 | 42 | 26 | 70 | 113 | No |
| 7 | M | 42 | 1b | No | G/P, 8 weeks | Yes | 3,300,000 | Not detected | 21 | 897 | 31 | 897 | 31 | No |
| 8 | F | 71 | 2 | No | G/P, 8 weeks | Yes | 780,000 | Not detected | 22 | 13 | 8 | 13 | 13 | No |
| 9 | M | 48 | 1b | Yes | IFN+RBV→DCV+ASV, 24 weeks | Yes | 1,300,000 | Not detected | 72 | 9,180 | 47 | >170,000,000 | 73 | Yes (TAF) |
| 10 | F | 73 | 1b | No | IFN+RBV→DCV+SOF, 12 weeks | Yes | 3,410,000 | 102 | 4 | 22,900 | 11 | 22,900 | 11 | No |
| 11 | M | 53 | 1b | No | IFN+RBV→OPr-D, 12 weeks | Yes | 452,000 | Not detected | 4 | 46 | 14 | 197 | 16 | No |
| 12 | F | 55 | 1b | Yes | IFN+RBV→DCV+ASV, 24 weeks | Yes | 3,580,000 | <20.0 | 16 | 693 | 24 | 1,010 | 42 | No |
| 13 | M | 51 | 1b | Yes | IFN+RBV→OPr-D, 12 weeks | Yes | 11,300,000 | Not detected | 48 | 267 | 23 | 267 | 68 | No |
| 14 | M | 49 | 1b | No | IFN+RBV→G/P, 8 weeks | Yes | 1,660,000 | Not detected | 10 | 4,580 | 16 | 4,580 | 16 | No |
| 15 | F | 66 | 1b | No | IFN+RBV→DCV+ASV, 10 weeks | No | 6,700,000 | Not detected | 62 | 114 | 76 | 114 | 76 | No |

HCV, hepatitis C virus; GT, genotype; LC, liver cirrhosis; SVR, sustained virological response; HBV, hepatitis B virus; ALT, alanine aminotransferase; NUC, nucleoside analogues; IFN, interferon; RBV, ribavirin; DCV, daclatasvir; ASV, asunaprevir; OPr-D, ombitavir/paritaprevir/ritonavir plus dasabuvir; SOF, sofosbuvir; G/P, glecaprevir/pibrentasvir; TAF, tenofovir alafenamide.

of an increase in HBV DNA up to 17,000,000 IU/mL and an increase in ALT to 73 U/mL, which did not meet the study's definition of HBV-related hepatitis.

**Factors associated with HBV reactivation among HBV/HCV-co-infected patients (Table 3)**

In univariate analysis, HBV reactivation was more common in the IFN-failed DAA-treated group than in the IFN-treated group (HR 8.750; 95% CI 1.397–54.799; p = 0.020). The HBV reactivation rate did not differ between the IFN-treated and DAA-treated groups or between the DAA-treated and IFN-failed DAA-treated groups. HBV reactivation was significantly more common in the non-HCV genotype 2 group in the univariate analysis (p = 0.043). In the multivariate analysis, HBV

reactivation was more frequent in the group receiving DAA treatment after the failure of IFN treatment (HR 10.767; 95% CI 1.014–114.362; p = 0.044).

## Safety and adverse events

There were no major clinical events related to HBV reactivation, such as liver decompensation or hepatic failure requiring liver transplantation.

## Discussion

This study identified HBV reactivations that occurred during or after three different HCV treatment regimens (IFN-based, DAA, and DAA after IFN-based treatment failure) in HBsAg-positive CHC patients during a median follow-up period of 53 months. The strengths of this study are that it included a large number of DAA-treated patients with prior IFN-based therapy, a population that has not been addressed in previous studies but is common in real-world practice, as well as the other two treatment groups, and followed them for a relatively long period of time.

In this study, the rates of HBV reactivation during or after HCV treatment in HBV/HCV coinfected patients were 31.9% in the overall cohort and 11.8%, 35.3%, and 53.8% in the IFN-based, DAA-treated, and DAA-treated patients after

**Table 3. Baseline features of patients according to the presence of HBV reactivation.** HBV, hepatitis B virus; HCV, hepatitis C virus; HCC, hepatocellular carcinoma; DAA, direct acting antiviral agents; DCV, daclatasvir; ASV, asunaprevir; SOF, sofosbuvir; RBV, ribavirin; OPr-D, ombitavir/paritaprevir/ritonavir plus dasabuvir; G/P, glecaprevir/pibrentasvir; LDV, ledipasvir; SVR, sustained virological response; IFN, interferon; DAA, direct acting antiviral.

| | HBV reactivation, No N = 32 | HBV reactivation, Yes N = 15 | Univariate P value | Multivariate P value |
|---|---|---|---|---|
| Age at HCV treatment (mean±SD) | 61.8 ± 9.5 | 58.4 ± 14.3 | 0.341 | |
| Male sex | 11 (31.4) | 9 (60.0) | 0.103 | |
| Presence of cirrhosis | 10 (31.3) | 5 (33.3) | 0.886 | |
| HCV genotypes<br>1<br>2<br>3<br>4<br>Unknown | 15 (46.9)<br>14 (43.8)<br>0 (0.0)<br>1 (3.1)<br>2 (6.3) | 12 (80.0)<br>2 (13.3)<br>1 (6.7)<br>0 (0.0)<br>0 (0.0) | 0.043 | 0.104 |
| History of HCC | 5 (15.6) | 0 (0.0) | 0.999 | |
| Treatment regimens of DAA (N = 30)<br>DCV+ASV<br>SOF+RBV<br>DCV+SOF<br>OPr-D<br>G/P<br>SOF+LDV | (N = 17)<br>3 (17.6)<br>6 (35.3)<br>0 (0.0)<br>1 (5.9)<br>5 (29.4)<br>2 (11.8) | (N = 13)<br>5 (38.5)<br>1 (7.7)<br>1 (7.7)<br>3 (23.1)<br>3 (23.1)<br>0 (0.0) | 0.077 | |
| Detectable HBV DNA before treatment | 12 (37.5) | 4 (26.7) | 0.467 | |
| SVR after antiviral treatment for HCV | 27 (84.4) | 14 (93.3) | 0.405 | |
| Treatment experience<br>IFN-based only<br>DAA only<br>DAA after treatment failure of IFN-based | 15 (46.9)<br>11 (34.4)<br>6 (18.8) | 2 (13.3)<br>6 (40.0)<br>7 (46.7) | 0.020 | 0.044* |
| Experience of specific therapy<br>IFN<br>DAA | 15 (46.9)<br>17 (53.1) | 2 (13.3)<br>13 (86.7) | 0.037 | |

IFN-based treatment failure arms, respectively. In a previous meta-analysis, the random effects pooled overall HBV reactivation rate was 15.7% in HBsAg-positive patients receiving anti-HCV treatment [33]. The higher overall HBV reactivation rate in this study compared with the meta-analysis findings may be due to the specific patient populations and treatment protocols used, including the presence of chronic HBV infection and previous IFN-based treatment failure. Other studies have reported HBV reactivation rates of 36.4% in the IFN-based treatment group [16,25,26] and 30.0%–57.0% in the DAA group [19,27–32], while in a meta-analysis of 1060 HBV/HCV-co-infected patients, the rates were 11.9% and 21.1% in the IFN-based and DAA treatment groups, respectively [33]. The incidence of HBV reactivation in the DAA group in this study was similar to that in previous studies [19,27–32], and the incidence of HBV reactivation in the IFN-based group was slightly lower than that in previous studies [16]. The similarity in the rate of HBV reactivation in patients treated with DAAs compared with those in previous studies suggests that the dynamic viral patterns and baseline HBV status of patients in this cohort are representative and not heterogeneous from those in cohorts in previous studies. HBV-related hepatitis, as defined in this study, occurred in only one patient and resolved without treatment 4 weeks after HBV reactivation. The incidence of HBV-related hepatitis in this study was significantly lower than that reported in meta-analyses that described a 9%−12% incidence of HBV-related hepatitis and liver failure requiring transplantation in the DAA-treated group, but similar to the findings in a retrospective study that found HBV reactivation in 9 of 62,290 DAA-treated patients in the United States [30,34,35].

To date, no study has statistically confirmed a difference in the frequency of HBV reactivation among the three different HCV treatments in HBV/HCV-co-infected patients, but we found that HBV reactivation was more common in the DAA after IFN-based treatment failure group than in the IFN-based treatment group. Another important feature of HBV reactivation in HBV/HCV-co-infected patients treated with DAAs identified in this study relates to the timing of HBV reactivation. In this study, HBV reactivation occurred significantly earlier in the DAA treatment group (4.2 months) and in the DAA after IFN treatment failure group (6.4 months) than in the IFN-based treatment group (44.5 months). These results are consistent with the meta-analysis findings describing that HBV reactivation occurred much earlier in the DAA treatment group, mostly 4–12 weeks during the DAA treatment, whereas HBV reactivation occurred at the end of IFN treatment or during the post-treatment follow-up period in the IFN-based treatment group [35].

It remains unclear what mechanisms underlie HBV reactivation after HCV treatment in patients with HBV/HCV co-infection and what accounts for the differences in the incidence and timing of HBV reactivation between IFN-based and DAA treatments. Studies have shown that de novo HCV superinfection in patients with CHB can lead to HBeAg seroconversion and, in some cases, the elimination of HBsAg, suggesting that HCV infection may suppress HBV replication in patients with HBV/HCV coinfection [36]. Therefore, since the suppression or elimination of HCV by hepatitis C treatment removes the inhibition of HBV replication, leading to HBV reactivation, it can be hypothesized that DAAs eliminate HCV more effectively and rapidly than IFNs, and thus the inhibitory effect of HCV on HBV replication is lost more quickly and more effectively, leading to more and faster HBV reactivation in HBV/HCV-co-infected patients. Unlike DAAs, IFN also exerts an inhibitory effect on HBV replication, and because it takes some time for the IFN effect to wear off after the EOT, this additional HBV-suppressive effect of IFN may ameliorate and delay HBV reactivation [37].

The SVR rate in the IFN-based treatment group was generally lower than that in the DAA therapy group. However, in our study a relatively high SVR rate of 82.4% was observed in the IFN-based treatment group. Several factors could contribute to this higher rate. The study was conducted over an extended period in a single center, which may have improved the response to IFN-based treatment. The long-term follow-up and single-center cohort characteristics could have influenced the patients' adherence to the treatment regimen, leading to a higher SVR rate. Additionally, high treatment compliance among the study participants is another crucial factor. Patients who are more compliant with their treatment regimen are more likely to achieve SVR, which could explain the higher rate observed in our study. The retrospective chart review methodology used in our study might also have contributed to the results. In summary, the higher SVR rate observed in the IFN-based treatment group in our study could be attributable to a combination of factors including cohort

characteristics, treatment compliance, study design, and methodology. These factors collectively contributed to a more favorable outcome for patients treated with IFN-based than typically observed in other studies.

The age at HCV treatment was lowest in the IFN-based treated group, followed by the DAA-treated group after IFN-based treatment failure, and then the DAA-treated group. This result is intriguing and warrants further exploration. For further clarification, it is important to note that the use of IFN-based treatments was often limited by their side effects. In patients with CHC, IFN-based therapy was generally recommended for patients with stage F2 or higher liver fibrosis and started as soon as possible for patients with advanced liver fibrosis (F3–4). In contrast, DAAs have been developed to be safer and more effective, allowing earlier treatment initiation and broader patient eligibility, including those with less severe disease. This selective treatment approach resulted in younger patients being more likely to receive IFN-HCV treatment, as they were more likely to have progressed to advanced fibrosis at a younger age. Conversely, the DAA-treated group included patients who were either treated earlier in the course of their disease or who experienced treatment failure with IFN, resulting in a higher median age than in the IFN-based treatment group.

Although several recently published sets of guidelines have recommended testing for HBV status before starting DAAs, whether to test for HBV DNA prior to DAA therapy and preemptive NUC treatment in HBV/HCV-co-infected patients remains controversial. This is because baseline HBV DNA levels before DAA treatment in HBV/HCV-co-infected patients are not predictive of HBV reactivation during treatment, and the cost of testing in HBV-endemic areas should be considered [35]. However, based on our findings, we suggest that, when planning DAA therapy, especially in patients with prior experience of IFN-based therapy failure, patients should be screened for evidence of HBV exposure, especially an HBsAg-positive status, and closely monitored for HBV reactivation from the start of DAA therapy until its end.

There is a need for further research on laboratory markers that predict HBV reactivation after HCV treatment in patients with HBV/HCV co-infection. Reactivation is known to occur when anti-HBs levels fall below 12 mIU/mL before and after DAA treatment [38,39], and studies have shown that high HBsAg titers before DAA treatment are associated with HBV reactivation [26]. Larger real-world studies using these markers are needed. HBV markers that predict DAA treatment-associated HBV reactivation and HBV-related hepatitis in HBV/HCV-co-infected patients require further investigation.

This study has several limitations. First, the retrospective study design has an inherent potential for selection bias. Second, this study is somewhat limited by its small sample size. To date, only five studies in populations with 100% HBsAg testing have been performed to the best of our knowledge, and the number of HBsAg-positive patients in our study is the third largest [26,28,40–42]. Third, this study did not identify laboratory test levels, such as quantifying pre-treatment HBsAg, that could predict HBV reactivation after DAA treatment in HBV/HCV-co-infected patients. Forth, the small numbers of patients and events may have limited the statistical power of our analysis. There is thus a need for further research in a larger cohort to confirm our findings. Fifth, our definition of HBV reactivation, which includes a 1 log increase in HBV DNA levels, may be more inclusive than some established criteria. While this lower threshold may lead to an overestimation of HBV reactivation rates, it aligns with the study's goal of examining the natural history of HBV reactivation in real-world clinical settings. By adopting a broader criterion, our approach also enables earlier detection of HBV reactivation, particularly in patients with advanced liver disease, which can facilitate timely intervention and potentially improve outcomes. Finally, the monitoring period in our study extended beyond the 12-week post-DAA treatment window recommended in some guidelines [18]. This broad timeframe may pose challenges when comparing results with studies that used stricter time limits, but it allowed us to capture HBV reactivation events occurring later in the course of treatment. In addition, another paper that observed HBV reactivation in CHC patients also observed follow-up for up to 48 weeks after completion of DAA treatment [43]. Current guidelines recommend maintaining prophylactic antiviral therapy for 6–12 months after completing treatments with high-risk drugs for HBV reactivation, suggesting that a longer monitoring period may be clinically relevant [44]. These limitations underscore the need for further research to establish standardized definitions and optimal monitoring durations for HBV reactivation in the context of DAA therapy for HCV, particularly given the potential for severe outcomes such as liver failure in some cases of HBV reactivation.

In conclusion, HBV reactivation in HBsAg-positive CHC patients is more common and occurs earlier in those who receive DAA treatment after IFN-based treatment failure than in IFN-based treatment. Therefore, all patients with CHC should be tested for HBV exposure prior to DAA treatment. In addition, HBsAg positive patients, especially those among whom have previously experienced IFN-based treatment failure, should be closely monitored for HBV reactivation during DAA therapy.

## Supporting information

**S1 Fig. Sustained virological response to antiviral therapy in HBV/HCV coinfected patients according to the DAA treatment regimen.** HBV, hepatitis B virus; HCV, hepatitis C virus; DAA, direct-acting antiviral; DCV, daclatasvir; ASV, asunaprevir; SOF, sofosbuvir; RBV, ribavirin; G/P, glecaprevir/pibrentasvir; OPr-D, ombitavir/paritaprevir/ritonavir plus dasabuvir; LDV, ledipasvir, SVR, sustained virological response.
(TIF)

**S2. Data set.**
(XLSX)

## Author contributions

**Conceptualization:** Hyun Young Woo, Jeong Heo.

**Data curation:** Young Joo Park, Ki Youn Yi.

**Formal analysis:** Young Joo Park.

**Supervision:** Hyun Young Woo, Jeong Heo, Geun Am Song.

**Writing – original draft:** Young Joo Park.

**Writing – review & editing:** Young Joo Park, Hyun Young Woo, Jeong Heo.

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
