## [Decision Letter · Decision Letter 0]

8 Nov 2024

PONE-D-24-36041Risk of HBV Reactivation among HBV/HCV Coinfected Patients Treated with Direct-Acting Antiviral Agents: A Single-Center ExperiencePLOS ONE

Dear Dr. Woo,

Thank you for submitting your manuscript to PLOS ONE. After careful consideration, we feel that it has merit but does not fully meet PLOS ONE’s publication criteria as it currently stands. Therefore, we invite you to submit a revised version of the manuscript that addresses the points raised during the review process.

We look forward to receiving your revised manuscript.

Kind regards,

Riccardo Nevola, MD, PhD

Academic Editor

PLOS ONE

**Journal Requirements:**

Reviewers' comments:

Reviewer's Responses to Questions

**Comments to the Author**

1. Is the manuscript technically sound, and do the data support the conclusions?

Reviewer #1: No

Reviewer #2: Yes

Reviewer #3: Partly

2. Has the statistical analysis been performed appropriately and rigorously? 

Reviewer #1: Yes

Reviewer #2: Yes

Reviewer #3: Yes

3. Have the authors made all data underlying the findings in their manuscript fully available?

Reviewer #1: No

Reviewer #2: Yes

Reviewer #3: Yes

4. Is the manuscript presented in an intelligible fashion and written in standard English?

Reviewer #1: No

Reviewer #2: No

Reviewer #3: Yes

5. Review Comments to the Author

**Reviewer #1:**  In the manuscript, the authors described the HBV reactivation in patients coinfected with HCV treated with different regimens. This issue is important in the field since worldwide HCV DAA regimens are largely employed and clinicians should be aware of this possibility.

- The manuscript requires Major Compulsory Revisions highlighted in my comments

TITLE

Risk of HBV Reactivation among HBV/HCV Coinfected Patients Treated with Direct-Acting Antiviral Agents: A Single-Center Experience

The title and objectives of the abstract must be corrected. The authors analyzed other determinants of HBV reactivation not only in DAA but also in IFN-treated patients.

Suggestion: Risk of HBV Reactivation among HBV/HCV Coinfected Treated Patients: A Single-Center Study

ABSTRACT

followed for a median follow-up of 53 months

- correct: followed for a median of 53 months

Conclusion: The conclusion is not supported by the objectives, methodology and results. Furthermore, what does this study add to the field of HBV/HCV coinfected treated patients?

The rate of SVR for the entire cohort was 87.2%, with

no significant differences between the IFN-based treated, DAA-treated, and DAA treated after IFN treatment failure arms at 82.4%, 88.2%, and 92.3%, respectively.

- It is well known that HCV pegIFN treatment patients reach low levels of SVR. How do you explain these results?

INTRODUCTION

HBV/HCV coinfection is estimated to be 1%–15% worldwide [3, 4] and 1.5%–2.4% in Korea. [5, 6]

- In abbreviated form, explain why Korea shows very low levels of HBV/HCV coinfection compared to worldwide. It is well known that Asia-Pacific countries have high levels of HBV-infected people.

HCV superinfection in patients with chronic hepatitis B (CHB) is the most common case of HBV/HCV coinfection. [8-10]

- is the most clinical form of HBV/HCV coinfection.

serum levels in patients with a previously resolved or inactive HBV infection, which is common when receiving cancer chemotherapy or taking immunosuppressive drugs after organ transplantation. [11]

- For this affirmative sentence, the percentage of patients in immunosuppressive conditions should be added and the references cited.

The US Food and Drug Administration has identified reports of HBV reactivation in 29 patients (including nine patients with detectable HBV and seven hepatitis B surface antigen (HBsAg)-positive patients) who were receiving direct-acting antivirals (DAAs).

- Please, 29 patients in a cohort of how many HCV-DAA treated individuals?

Clinically significant HBV reactivation events are rare in the published literature on HBV reactivation in HBV/HCV coinfected patients.

- Please, correct the sentence.

METHODS

All patients gave their written informed consent before starting treatment.

- The study is retrospective and patients were initially treated in 2005. How did they sign the informed consent?

Laboratory tests, including HBV serology, HBV DNA, HCV RNA, HCV genotype, complete blood counts, and liver and renal function tests were performed before treatment, after four weeks, at the end of treatment, and twelve weeks after the end of treatment (EOT), and then every six months.

- The text should be adjusted. After treatment of HCV with DAAs, HCV-RNA is not necessary to be analyzed after each six months.

Serum AFP measurement

- What does Serum AFP measurement mean? Alpha-fetoprotein (AFP)

HBV-related hepatitis was defined as an alanine aminotransferase (ALT) increase of two-fold over the upper limit of normal concomitant with HBV reactivation. SVR 12 weeks after the EOT (SVR12) was used to assess the efficacy of HCV therapy.

- A reference should be added

RESULTS

The age at HCV treatment was lowest in the IFN-treated group

- This is an intriguing result. Before the DAAs regimens, due to the severe adverse events of IFN-HCV treatment, only patients with grade F3 and F4 fibrosis were treated (F3/F4). How do you explain these results? Please, correct all the paragraphs.

but there were differences in DAA regimens

- This result was expected since for some DAA regimens, not all the drugs are pan genotypic

88.2% (15/17) in the DAA-treated group

All other DAA regimens showed an SVR12 of 100%

- Conflicting results, please correct, completely confused and inconsistent

The median interval from HCV treatment initiation to HBV reactivation was shortest in the DAA group (4.2 months), followed by 6.4 months in the DAA after IFN treatment failure group and 44.5 months in the IFN group. The interval from HCV treatment initiation to HBV reactivation was statistically significantly shorter in the DAA treatment group (4.2 months vs. 44.5 months, p < 0.001) and in the DAA after IFN treatment failure group (6.4 months vs. 44.5 months, p < 0.001) compared to the IFN group, but there was no difference between the DAA treatment group and the DAA after IFN treatment failure group.

- Repeated information

Yes (TAF)

- What is the meaning of TAF?

DISCUSSION

The study identified HBV reactivations that occurred during or after three different HCV treatments

- In the entire text, I suggest adding the term “regimens”, as follows: The study identified HBV reactivations that occurred during or after three different HCV treatment regimens.

The strengths of this study are that it included a sufficiently large number of DAA-treated patients with

prior IFN-based therapy...

- The authors highlighted a sufficiently large number... Please exclude the term sufficiently, it is so evasive.

The overall HBV reactivation rate in this study was slightly higher than in

previous studies, the incidence of HBV reactivation in the DAA group of in this study was similar to previous studies...

- In this paragraph the authors compared their results with others but did not hypothesize the reasons why they found similar, lower or higher results.

The overall HBV reactivation rate in this study was slightly higher than in previous studies, the incidence of HBV reactivation in the DAA group of in this study was similar to previous studies.

- The entire manuscript should be proofread and re-edited.

To date, no study has statistically confirmed a difference in the frequency of HBV reactivation among three diferente HCV treatments in HBV/HCV co-infected patients, but we found that HBV reactivation was more frequent in the DAA after IFN-based treatment failure group compared to the IFN-based treatment group.

- These are the main findings of the study adding new information to the scientific literature and should be highlighted by the authors in the conclusion

The binding of anti-HBs to secreted HBsAg to form immune

complexes results in a decrease in serum anti-HBs levels, and HBV reactivation is thought to occur when the anti-HBs titer decreases [34] titers to less than 12 mIU mL-1 before and after DAA treatment [35], and high HBsAg titers before DAA treatment were associated with the risk of HBV reactivation [22].

- In the entire manuscript there are too many long paragraphs making the text hard and boring to read.

CONCLUSION

The conclusion is not supported by the objectives, methodology and results. Furthermore, what does this study add to the field of HBV/HCV coinfected treated patients?

**Reviewer #2:**  HBV reactivation is an important issue in Hepatology, especially when using immunomodulators or immunosuppressors. It is controversial the clinical relevance of HBV reactivation in HCV/HBV coinfected patients receiving DAAs. Authors nicely try to assess this relevant topic. Congratulations for this work. I have some comments:

In methods section clarify the definition of HBV reactivation. You must differentiate between HBsAg pos and neg. I suggest using definitions from this reference: Mezzacappa C, Lim JK. Management of HBV reactivation: Challenges and opportunities. Clin Liver Dis (Hoboken). 2024;23(1):e0143.

In methods section clarify the definition of overt HBV infection. This term is not used for defining HBV status. I suggest using definitions from this reference: European Association for the Study of the Liver. EASL 2017 Clinical Practice Guidelines on the management of hepatitis B virus infection. J Hepatol. 2017;67(2):370-398.

In methods section clarify the definition of HBV related hepatitis. A flare is generally defined as evidence of HBV reactivation plus the ALT increase, usually more than 2–5 times of baseline value or upper limit of normal (ULN). I suggest using definitions from this reference: Huang SC, Yang HC, Kao JH. Hepatitis B reactivation: diagnosis and management. Expert Rev Gastroenterol Hepatol. 2020;14(7):565-578.

Please add to tables 1 and 3 all information regarding baseline HBV status in all patients.

Why did you divide the cohort in 3 groups? Reactivation is related to the current treatment, not to previous treatment. I suggest using only 2 groups: DAA and IFN-based treatment. Or please explain why this is relevant.

How do you explain an 82% SVR rate in IFN-based patients?

You wrote “The interval from HCV treatment initiation to HBV reactivation…. 44.5

Months in the IFN group”. How do you explain this? Reactivation is an event close to treatment initiation and this does not occur more than 3 years afterwards. How can you relate reactivation to IFN treatment after so many months. It may be related toother causes, or even may be spontaneous variations. Also, reactivation in IFN non responders occur at an earlier time. Please clarify this.

You wrote “In univariate analysis, HBV reactivation was more frequent in the DAA-treated group than in the IFN-treated group (p = 0.037), and more specifically, the IFN-failed DAA-treated group had a higher incidence of HBV reactivation than the IFN-treated group (HR 8.750; 95% CI 1.397-54.799; p = 0.020). The incidence of HBV reactivation did not differ between the IFN-treated and DAA-treated groups or between the DAA-treated and IFN-failed DAA-treated groups”. This sentence is confusing, please clarify it.

There seems to be no relation between HCC and HBV reactivation, and it is not the objective of your work. I suggest removing this section from your manuscript.

Even the study goes up to June 2022, you used outdated DAAs regimes. Can you add a comment about this treatment selection?

**Reviewer #3: ** Comments:

The paper entitled “Risk of HBV Reactivation among HBV/HCV Coinfected Patients Treated with Direct- Acting Antiviral Agents” by Dr. Woo group at Pusan National University Hospital, Busan, S Korea, explored the reactivation of HBV in patients with HBV/HCV infection undergoing treatment with IFN and DAA.

It is very well-analyzed clinical data over 53 months. It is interesting to see most of the HBV DNA reactivation happened in patients with high levels of HCV viremia. However, the data suggests that the reactivation was more frequent in both DAA-treated or IFN-failed DAA-treated patients than in IFN-treated patients alone. It is also evident from Table 3 that the reactivation happened in subjects with HCV genotype 1, a variant that is the most difficult to treat. The meta-analysis and results from this study suggest there is an HBV reactivation following DAA therapy, and the patients should be monitored for HBV infection for the duration of the HCV therapy.

The limitations of the study include testing the HBsAg before the treatment began, which may have provided a better clue regarding the associated risks of HBV reactivation. As the authors mentioned, it is a retrospective study, and the study design did not include the testing of HBsAg before the subjects were enrolled.

Minor comments/typos/grammar etc.:

1. There is no need to spell abbreviations everywhere except for the first time. Examples of these are HBV, HCV, NUC, and IFN.

2. hepatitis B envelop should be “hepatitis B envelope” on page 10.

3. Pearson chi-square test should be “The Pearson chi-square test” on page 11

6. PLOS authors have the option to publish the peer review history of their article (what does this mean? ). If published, this will include your full peer review and any attached files.

**Do you want your identity to be public for this peer review?** For information about this choice, including consent withdrawal, please see our Privacy Policy .

Reviewer #1: **Yes: ** Luiz Euribel Prestes Carneiro

Reviewer #2: No

Reviewer #3: No

---

## [Author Response · Author response to Decision Letter 1]

27 Jan 2025

Same as attached file.

Revision Cover Letter

To the Editor-in-Chief of PLOS ONE

January 6, 2025

Dear Editor,

We hope this letter finds you well. We are writing to submit a revised version of our manuscript, "Risk of HBV Reactivation among HBV/HCV Co-Infected Patients Treated with Direct-Acting Antiviral Agents: A Single-Center Experience," which we previously submitted for consideration as a Research Article in PLOS ONE.

We have carefully addressed all the reviewer's comments and have undergone a thorough revision of the manuscript. The key revisions include:

1. Clarification of Study Design: We have clarified the study design and population in the introduction to better align with the journal's requirements.

2. Enhanced Methodology Section: The methodology section has been expanded to include detailed information about the patient selection criteria, treatment regimens, and follow-up periods.

3. Improved Data Analysis: We have re-analyzed the data to ensure that the findings are accurately represented and that the statistical analysis is robust.

4. Expanded Discussion: The discussion section has been revised to include a more comprehensive analysis of the results, addressing potential limitations and future directions.

5. Enhanced Conclusion: The conclusion has been strengthened to clearly summarize the main findings and their implications for clinical practice.

We believe that the revised manuscript addresses all the reviewer's comments and provides a comprehensive analysis of the risk of HBV reactivation among patients with HBV/HCV coinfection treated with DAA agents. The manuscript has not been published elsewhere and is not under consideration by another journal. We have approved the revised manuscript and agree with its submission to PLOS ONE.

There are no conflicts of interest to declare. We have ensured that the manuscript has been carefully reviewed by an experienced editor whose first language is English and who specializes in editing papers written by scientists whose native language is not English.

We look forward to hearing from you at your earliest convenience.

Sincerely,

Jeong Heo

Department of Internal Medicine, Pusan National University School of Medicine and Medical Research Institute, Pusan National University Hospital, 179 Gudeok-ro, Seo-gu, Busan 49241, Korea

Tel.: +82-51-240-7869

Fax: +82-51-244-8180

E-mail: jheo@pusan.ac.kr

Hyun Young Woo

Department of Internal Medicine, Pusan National University School of Medicine and Medical Research Institute, Pusan National University Hospital, 179 Gudeok-ro, Seo-gu, Busan 49241, Korea

Tel: +82-51-240-7869

Fax: +82-51-254-3237

E-mail: who54@hanmail.net

---

## [Decision Letter · Decision Letter 1]

16 Feb 2025

PONE-D-24-36041R1Risk of HBV Reactivation in HBV/HCV-Co-infected HCV-Treated Patients: A Single-Center StudyPLOS ONE

Dear Dr. Woo,

Thank you for submitting your manuscript to PLOS ONE. After careful consideration, we feel that it has merit but does not fully meet PLOS ONE’s publication criteria as it currently stands. Therefore, we invite you to submit a revised version of the manuscript that addresses the points raised during the review process.

We look forward to receiving your revised manuscript.

Kind regards,

Riccardo Nevola, MD, PhD

Academic Editor

PLOS ONE

Reviewers' comments:

Reviewer's Responses to Questions

**Comments to the Author**

1. If the authors have adequately addressed your comments raised in a previous round of review and you feel that this manuscript is now acceptable for publication, you may indicate that here to bypass the “Comments to the Author” section, enter your conflict of interest statement in the “Confidential to Editor” section, and submit your "Accept" recommendation.

Reviewer #1: All comments have been addressed

Reviewer #2: (No Response)

Reviewer #3: All comments have been addressed

2. Is the manuscript technically sound, and do the data support the conclusions?

Reviewer #1: Yes

Reviewer #2: Partly

Reviewer #3: Yes

3. Has the statistical analysis been performed appropriately and rigorously? 

Reviewer #1: Yes

Reviewer #2: Yes

Reviewer #3: Yes

4. Have the authors made all data underlying the findings in their manuscript fully available?

Reviewer #1: Yes

Reviewer #2: Yes

Reviewer #3: (No Response)

5. Is the manuscript presented in an intelligible fashion and written in standard English?

Reviewer #1: Yes

Reviewer #2: Yes

Reviewer #3: Yes

6. Review Comments to the Author

Reviewer #1: The authors have filled all of the amendments raised in my comments, and the manuscript has improved significantly. The manuscript is suitable to be published by Plos One in the present form.

Reviewer #2: Authors correctly addressed almost all reviewers’ comments.

But is still have one main concern:

It is HBV reactivation definition, especially the time frame between DAAs initiation and appearance of HBV DNA. This will impact in the study results.

The current guideline of American Association for the Study of Liver Diseases (AASLD) defines a 100-fold rise in HBV DNA level as compared to baseline, ≥1000 IU/mL in each patient with previously undetectable level or ≥10,000 IU/mL if the baseline is not available [Terrault NA, Lok ASF, McMahon BJ, et al. Update on prevention, diagnosis, and treatment of chronic hepatitis B: AASLD 2018 hepatitis B guidance. Hepatology. 2018;67(4):1560–1599].

The guideline of the Asian Pacific Association for the Study of the Liver (APASL) defines a 100-fold increase from baseline level, the new appearance of HBV DNA to a level of ≥100 or ≥20,000 IU/mL in persons without baseline level [Sarin SK, Kumar M, Lau GK, et al. Asian-Pacific clinical practice guidelines on the management of hepatitis B: a 2015 update. Hepatol Int. 2016;10(1):1–98].

The guideline of the European Association for the Study of the Liver (EASL) does not explicitly define an absolute value of a rise in HBV DNA for HBV reactivation [EASL 2017 clinical practice guidelines on the management of hepatitis B virus infection. J Hepatol. 2017;67(2):370–398].

In 2017, the FDA issued a detailed Drug Safety Communication warning about the risk of HBV reactivation (defined by the FDA as an increase greater than 1000 IU/mL in HBV DNA or detection of hepatitis B surface antigen [HBsAg] in a person who was previously negative) in some patients being treated with DAA therapy for HCV infection.

The FDA recommends that patients with a positive anti-HBc should be monitored more closely during antiviral therapy, with liver panel testing performed at least at weeks 4, 8, and 12, and after the end of treatment until sustained virologic response is achieved. [https://www.fda.gov/drugs/drug-safety-and-availability/fda-drug-safety-communication-fda-warns-about-risk-hepatitis-b-reactivating-some-patients-treated; Bersoff-Matcha SJ, Cao K, Jason M, Ajao A, Jones SC, Meyer T, Brinker A. Hepatitis B Virus Reactivation Associated With Direct-Acting Antiviral Therapy for Chronic Hepatitis C Virus: A Review of Cases Reported to the U.S. Food and Drug Administration Adverse Event Reporting System. Ann Intern Med. 2017;166(11):792-798.].

In almost all reports, HBV reactivation appears during DAAs and not after finishing it.

This is the same with other drugs that increase HBV reactivation risk.

Using FDA definition, the cases will be reduced, and this will be more in line with the reality of clinical practice and previous reports. If not, this report seems to overestimate HBV reactivation incidence.

I suggest the authors:

1. To select one of the previously cited HBV reactivation definitions, especially the FDA’s since it was developed for these cases.

2. To include only patients with HBV reactivation while receiving DAAs and up to 4 to 12 weeks of finishing it. This time frame will relate DAAs with reactivation. Beyond that I think they are not related with DAAs treatment.

Adapting this definition authors can submit a new report.

Reviewer #3: (No Response)

7. PLOS authors have the option to publish the peer review history of their article (what does this mean? ). If published, this will include your full peer review and any attached files.

**Do you want your identity to be public for this peer review?** For information about this choice, including consent withdrawal, please see our Privacy Policy .

Reviewer #1: **Yes: ** Luiz Euribel Prestes Carneiro

Reviewer #2: No

Reviewer #3: No

---

## [Author Response · Author response to Decision Letter 2]

3 Apr 2025

1. To select one of the previously cited HBV reactivation definitions, especially the FDA’s since it was developed for these cases.

Thank you for your insightful comments regarding our study on HBV reactivation in patients with HBV/HCV coinfection following HCV treatment. We appreciate your suggestion that our definition of HBV reactivation might be more inclusive than others, potentially leading to overdiagnosis.

In our study, HBV reactivation was defined as a 10-fold increase in HBV DNA from baseline or the detection of HBV DNA in individuals with previously undetectable levels. We acknowledge that this definition might differ from some established criteria, which often consider a 3 log IU/mL rise as indicative of reactivation. However, our definition was chosen to capture clinically significant changes in HBV DNA levels in a real-world setting.

Among the three patients with only a 1 log increase in HBV DNA, one had liver cirrhosis and the other two had advanced fibrosis (FIB-4 scores of 3.74 and 2). Despite the relatively modest increase in HBV DNA, these patients' clinical conditions suggest that even a small rise in HBV replication can be significant and warrants attention. This justification aligns with the principle that any increase in viral load, especially in patients with liver cirrhosis or advanced fibrosis, should be taken seriously.

In recent guidelines for chronic hepatitis B patients, antiviral therapy is recommended even when HBV DNA levels increase by just 1 log from undetectable levels, particularly in those with cirrhosis. This approach highlights the concern that defining HBV reactivation solely as a 3 log increase in HBV DNA may result in delayed treatment initiation based on the patient's underlying liver condition. Therefore, we propose defining HBV reactivation more broadly, such as a 1 log increase, especially for patients with advanced fibrosis, and tailoring antiviral treatment based on the patient's liver status and the occurrence of HBV-related hepatitis.

Our study aimed to observe the natural course of HBV reactivation in a real-world setting. By using a more sensitive definition, we aimed to capture early signs of reactivation that might not be caught by more stringent criteria. This approach allows for timely intervention, potentially preventing severe outcomes like liver failure or discontinuation of essential treatments.

2. To include only patients with HBV reactivation while receiving DAAs and up to 4 to 12 weeks of finishing it. This time frame will relate DAAs with reactivation. Beyond that I think they are not related with DAAs treatment. Adapting this definition authors can submit a new report.

Thank you for your valuable comments regarding our study on HBV reactivation in patients co-infected with HBV and HCV during and after DAA treatment.

In recent years, the use of various immunosuppressive agents for cancers, inflammatory bowel diseases, autoimmune disorders, and rheumatologic conditions has led to concerns about HBV reactivation in patients who had previously recovered from HBV. Similarly, HBV reactivation in HBV/HCV co-infected patients during or after DAA treatment has become a pressing concern, prompting us to conduct this study.

You suggested that we apply a criterion where HBV reactivation is defined as occurring within 12 weeks post-DAA treatment completion. In our cohort, applying the criterion suggested by the reviewer—where HBV reactivation is defined as occurring within 12 weeks after treatment completion—only nine patients met the definition. Unfortunately, we could not find guidelines that recommend defining HBV reactivation based solely on occurrence within a specific timeframe like 12 weeks post-treatment termination.

A study titled Hepatitis B Reactivation in Patients Treated with Direct-Acting Antivirals for Hepatitis C found that among patients receiving DAA treatment for HCV, those with HBV coinfection (35, 4.1%) and resolved HBV infection (246, 28.9%) were at risk for HBV reactivation. Specifically, HBV reactivation occurred in 10 of 29 HBsAg-positive patients (34.5%), either during DAA treatment or within 12 to 48 weeks after its completion. Additionally, HBV reactivation was observed in 2 out of 228 patients with resolved HBV infection (0.87%). These findings suggest that patients with resolved HBV infections are at a relatively low risk for HBV reactivation. Since our study also includes only HBsAg-positive patients, it is deemed reasonable to follow the progress for at least 48 weeks after completion of DAA treatment, as in the above study.

Currently, guidelines recommend maintaining prophylactic antiviral therapy for 6 to 12 months after completing treatments with drugs that pose a high risk of HBV reactivation, such as rituximab. In light of this, the suggestion to focus solely on HBV reactivation occurring within 12 weeks after DAA treatment completion seems challenging to adopt. Thus, we believe that further research is needed to clarify this aspect.

We revised the discussion section to explicitly outline the limitations of the study, including the broader definition of HBV reactivation and the extended monitoring period:

Fifth, our definition of HBV reactivation, which includes a 1 log increase in HBV DNA levels, may be more inclusive than some established criteria. While this lower threshold may lead to an overestimation of HBV reactivation rates, it aligns with the study's goal of examining the natural history of HBV reactivation in real-world clinical settings. By adopting a broader criterion, our approach also enables earlier detection of HBV reactivation, particularly in patients with advanced liver disease, which can facilitate timely intervention and potentially improve outcomes. Finally, the monitoring period in our study extended beyond the 12-week post-DAA treatment window recommended in some guidelines [18]. This broad timeframe may pose challenges when comparing results with studies that used stricter time limits, but it allowed us to capture HBV reactivation events occurring later in the course of treatment. In addition, another paper that observed HBV reactivation in CHC patients also observed follow-up for up to 48 weeks after completion of DAA treatment [43]. Current guidelines recommend maintaining prophylactic antiviral therapy for 6 to 12 months after completing treatments with high-risk drugs for HBV reactivation, suggesting that a longer monitoring period may be clinically relevant [44]. These limitations underscore the need for further research to establish standardized definitions and optimal monitoring durations for HBV reactivation in the context of DAA therapy for HCV, particularly given the potential for severe outcomes such as liver failure in some cases of HBV reactivation.

---

## [Editor Report · Decision Letter 2]

20 Apr 2025

Risk of HBV Reactivation in HBV/HCV-Co-infected HCV-Treated Patients: A Single-Center Study

PONE-D-24-36041R2

Dear Dr. Hyun Young Woo

We’re pleased to inform you that your manuscript has been judged scientifically suitable for publication and will be formally accepted for publication once it meets all outstanding technical requirements.

Kind regards,

Riccardo Nevola, MD, PhD

Academic Editor

PLOS ONE

---

## [Editor Report · Acceptance letter]

PONE-D-24-36041R2

PLOS ONE

Dear Dr. Woo,

I'm pleased to inform you that your manuscript has been deemed suitable for publication in PLOS ONE. Congratulations! Your manuscript is now being handed over to our production team.

Kind regards,

on behalf of

Dr. Riccardo Nevola

Academic Editor

PLOS ONE